# A Study of Varicella Seroprevalence in a Pediatric and Adolescent Population in Florence (Italy). Natural Infection and Vaccination-Acquired Immunization

**DOI:** 10.3390/vaccines9020152

**Published:** 2021-02-14

**Authors:** Beatrice Zanella, Angela Bechini, Benedetta Bonito, Marco Del Riccio, Alessandra Ninci, Emilia Tiscione, Paolo Bonanni, Sara Boccalini

**Affiliations:** 1Department of Health Sciences, University of Florence, 50134 Florence, Italy; beatrice.zanella@unifi.it (B.Z.); benedetta.bonito@unifi.it (B.B.); emilia.tiscione@unifi.it (E.T.); paolo.bonanni@unifi.it (P.B.); sara.boccalini@unifi.it (S.B.); 2Medical Specialization School of Hygiene and Preventive Medicine, University of Florence, 50134 Florence, Italy; marco.delriccio@unifi.it (M.D.R.); alessandra.ninci@unifi.it (A.N.); gino.sartor@unifi.it (W.G.D.); 3Meyer Children’s Hospital, 50139 Florence, Italy; francesco.puggelli@meyer.it; 4AUSL Toscana Centro, 50122 Florence, Italy; giovanna.mereu@uslcentro.toscana.it

**Keywords:** varicella, vaccination coverage, immunization, notification, Italy, pediatric, adolescent, seroprevalence, Tuscany, underreporting

## Abstract

*Background:* Varicella is a well-known infectious disease that can have severe complications, also in young children. The Universal Varicella Vaccination (UVV) program was introduced in Tuscany (Italy) in 2003, with a two-dose vaccine schedule given to children between their 13th and 15th month, and at 5–6 years old, as a monovalent for varicella (V) or tetravalent (measles, mumps, rubella and varicella (MMRV)) formulation. Although varicella notifications have dramatically fallen in the last two decades, varicella disease underreporting remains a challenge. *Methods*: A qualitative immunoenzymatic test (ELISA) was used to measure the presence of anti-varicella antibodies in 165 sera of subjects aged 1–18 years residing in the province of Florence (Italy). Information regarding the anamnestic and vaccination status (including disease notification) was also collected. *Results:* Our study showed an overall varicella seropositivity of 75.8% (reaching the maximum at 96.3% in the 15–18 years age group). We found that varicella disease notification had been recorded for only 7/165 subjects; however, since 42/165 recalled having had the disease, we can hypothesize that some of them must have been underreported. Furthermore, our study showed that the presence of antibodies after the varicella vaccination remained over time, lasting up to 12 years. *Conclusions:* Although varicella seroprevalence is <95% in almost all our age groups (except for the 15–18 years age group), our data are encouraging and reflect the success of the introduction of the UVV program and the vaccination campaigns promoted in the Tuscany region.

## 1. Introduction

Varicella disease (chickenpox) is caused by a highly contagious virus named varicella zoster virus (VZV). Varicella is the primary infection of VZV, and it can be followed by a secondary infection (zoster or shingles). The main way of transmission is either by inhalation of aerosols droplets from vesicular fluid of the lesions or by direct contact with the skin rash of an infected person (unvaccinated people can have around 300 lesions). Disease incubation time is usually 14–16 days [1]. Varicella symptoms are fever, malaise and vesicular rash generally concentrated in the neck and trunk. Generally, it is not a severe or self-limiting disease. However, a few complications may occur, including bacterial superinfections (cellulitis, pneumonia, osteomyelitis) and neurological complications (cerebellar ataxia, encephalitis) [2]. These complications often occur in people older than 15 years, people younger than 1 year, those who are immunocompromised and newborns from women who have had a rash during the period between 5 days before and 2 days after delivery [3]. In particular, varicella in pregnant women may be transmitted to the fetus and, depending on the time of infection, may result in intrauterine death and congenital varicella syndrome (CVS), characterized by various fetal abnormalities [4]. After the primary infection, VZV remains latent in the dorsal and cranial root ganglia and may be reactivated by certain conditions leading to virus replication, inflammation and cell death [5]. In the prevaccine era of the United States of America, almost 11,000 varicella-affected people were hospitalized each year (~3/1000 cases amongst healthy children and 8/1000 cases amongst adults); deaths occurred in approximately 1/60,000 cases, mostly in immunocompetent children and adults [3]. However, when the universal live-attenuated, one-dose vaccination was introduced in the United States in 1995, the number of hospitalizations and deaths from varicella dropped by ~70% and ~88%, respectively [3,6]. However, following a “breakthrough varicella” (BV)—defined as a case of infection with wild-type VZV occurring >42 days after vaccination—the Advisory Committee on Immunization Practices (ACIP) recommended a second vaccination dose for children aged 4–6 years in 2007 [7]. The universal varicella vaccination (UVV) for children in the EU/EEA (European Union/European Economic Area) is recommended at a national level in Hungary, Iceland, Cyprus, Germany, Greece, Latvia, Luxembourg, Austria, Italy [8,9,10], Finland [11] and, more recently, Spain [12]. Moreover, 17 European countries endorsed nationwide vaccine administration for susceptible teenagers and/or susceptible risk groups only (i.e., healthcare workers) [9]. A two-dose measles, mumps, rubella and varicella (MMRV) vaccination schedule is generally administered around the ages of 13–15 months and 5–6 years in most countries in Europe [13]. In Italy, UVV was originally introduced in 2003, and it was initially limited to the regions able to guarantee a high level (>80%) of vaccination coverage (VC), including the Tuscany region in 2008 [14]. This program has since been expanded to the whole country with its National Immunization Plan (NIP) 2017–2019 [15]. Hence, the number of varicella cases in Italy has dropped from ~10,000 cases in 2003 to ~4000 in 2018 [16]. Mirroring the European scenario, the Italian vaccination protocol recommends a two-dose administration schedule: between the 13th and 15th months, and at 5–6 years, administered in a monovalent (V) or tetravalent (MMRV) formulation. The vaccination is also offered free of charge to susceptible groups: particularly adults and adolescents, where two doses across 4 weeks are administered. Even though the number of varicella notifications in Tuscany has dramatically fallen in the last decades, the notification rate remains high (from ~350 cases/100,000 in 1994 to 20 cases/100,000 in 2018—i.e., >700 notifications in 2018) [17]. In the same region, the disease is contracted typically between the ages of 5 and 14 years, with a median age of 12 years in 2017, which has increased compared to previous years [17]. Although varicella is a compulsory notifiable disease, the underreporting rates are still high, with considerable differences between Italian regions [18]. In order to improve the notification surveillance system, a Pediatric Sentinel Surveillance System provided by Italian pediatricians (Sistema Pediatrico di sorveglianza dei Pediatri Sentinella (SPES)) was established in January 2000 [17]. 

Varicella seroprevalence investigation is part of a wider seroepidemiological project that has been carried out by the Department of Health Sciences of the University of Florence since 2017, aiming to assess varicella seroprevalence in the same sample population used for measles [19], rubella [20], hepatitis A and B [21] and tetanus surveillance. The overall goal of this study was to investigate the varicella seroprevalence in a pediatric and adolescent (1–18 years) population of the province of Florence and to relate the obtained data with the anamnestic and vaccination status of each subject, with particular attention to the varicella notifications.

## 2. Materials and Methods 

### 2.1. Subject Enrolment and Sera Collection 

The study was conducted in accordance with the Declaration of Helsinki, and the protocol was approved by the local ethics committee (project identification code: DSS-UNIFI, *n. registro pareri* 98/2017). Estimating an anti-varicella seroprevalence of 81% (with an accuracy of 6% and a confidence level of 95%), the size of the sample population was 164 sera, calculated in a post hoc analysis. The number of sera approximately represented 0.1% of the resident population of Florence aged 1–18 years in a total of 166,644 subjects in 2017 in the same age group [22]. This number is also proportionally related to the population composition for each age group and gender. Therefore, further standardization was not necessary. Subjects who were nonresident in the province of Florence, immunocompromised patients or patients under immunosuppressive treatment, subjects who had an acute infectious disease (among measles, rubella, varicella, hepatitis A and hepatitis B) in the previous 2 weeks and subjects who had received a blood transfusion within 6 months before enrolment into the study were excluded. At the time of their recruitment, the subjects’ parents or guardians gave written consent to be included in the study. The enrolment and the collection of the sera took place at the blood sampling center of the Meyer Children’s Hospital, starting in December 2017 and finishing in April 2018.

### 2.2. Anamnestic and Vaccination Status

The anamnestic status of each subject was retrieved using The National Registry of Notifications for Infectious Diseases (SIMI, software: Epi Info, Rome, Italy). We discovered the vaccination status for varicella for each subject through the vaccination register’s Collective Prevention Healthcare Information System (SISPC; Consortium Metis, Tuscany, Italy) and Caribel (Aster, Tuscany, Italy), the current and the previous VC software used in the Tuscany region, respectively. The number of varicella vaccine doses, the year of the last dose and, if available, the type of the last administered vaccine were retrieved. At enrolment, a “disease recall section” in the questionnaire allowed the subject to recall whether they had had the disease, even if it had not been officially reported. 

### 2.3. Serological Analysis 

A qualitative measurement of anti-varicella antibodies was obtained using the commercial varicella anti-virus/IgG ELISA kit (Enzygnost). All the collected sera were tested for anti-varicella IgG antibodies, and each result was classified according to the following absorbance set values: Anti-varicella/IgG negative ΔA < 0.100 (cut-off);Anti-varicella/IgG positive ΔA > 0.200;Anti-varicella/IgG equivocal 0.100 ≤ ΔA ≤ 0.200.

The ΔA value was calculated as the difference between the absorbance obtained for each sample and the absorbance value of the same sample containing the control varicella antigen (provided by the manufacturer). In the case of an equivocal outcome, the sample was analyzed a second time. If the result was still equivocal, the outcome of that sample was then classified as equivocal.

### 2.4. Statistical Analysis 

These qualitative varicella data were collated in an Excel database and analyzed in terms of the following factors: antibodies present (positive, negative or equivocal), gender, nationality, age group, vaccination status, number of vaccine doses received, and the time elapsed since the last dose of vaccine received. For the analysis, all the subjects with an Italian nationality and all the subjects with a foreign nationality either born in Italy or adopted were classified as Italian, whereas subjects born abroad with a double nationality (both Italian and foreign) or a foreign nationality were classified as non-Italian.

Differences between sex, age, nationality, vaccination status, number of doses of vaccine and time elapsed since the last dose of vaccine were evaluated. Statistical significance was assessed with the two-tailed chi-squared test or Fisher’s exact test (assuming a *p*-value < 0.05 as statistically significant). A Cochran–Armitage test (CAT) for trends was used to assess the presence of linear trends between serological status and ordinal variables (age categories and years elapsed from the last dose of vaccine), assuming as nonpositive the subjects found to be negative or equivocal. The statistical analyses were conducted using RStudio 1.2.5033 (RStudio Team, 2019. RStudio: Integrated Development for R. RStudio, Inc., Boston, MA; URL: http://www.rstudio.com; accessed on 20 January 2021).

Our sample population was divided into four age groups: 40 subjects in the 1–4 years age group, 48 in the 5–9 years age group, 50 in the 10–14 years age group and 27 in the 15–18 years age group. Males were 53.3% of the study population, and females were 46.7%. Most of the enrolled subjects were Italian citizens (90.3%), and the remaining part (9.7%) consisted of non-Italian subjects (dual nationality or foreign). The participants resided in 35 different districts of the Province of Florence, and almost half of them (48%) were living in the City of Florence.

## 3. Data Analysis

### 3.1. Varicella Seroprevalence Analysis

The overall seroprevalence against anti-VZV antibodies was 75.8%, while 13.3% of the enrolled subjects were negative and 10.9% were equivocal. The seroprevalence distribution was similar among male and female subjects and among Italian and non-Italian subjects, as described in Table 1. 

No significant difference among the seropositive subjects was found between male and female subjects (*p* > 0.05) or between Italian and non-Italian subjects (*p* > 0.05). 

Seropositivity was comparable in the age groups 1–4 years (65%, 26/40) and 5–9 years (68.8%, 33/48), while it increased in the older age groups up to 80% (40/50) in subjects aged 10–14 years old and up to 96.3% (26/27) in adolescents aged 15–18 years (with no equivocal samples found in this category), showing a significant increase in the number of seropositive subjects while increasing age (*p* < 0.01) (Figure 1).

### 3.2. Varicella Disease Notification and Subjects’ Anamnestic Recall of the Disease 

According to SIMI, varicella notification had been recorded for seven subjects, and five of them experienced the infectious disease at approximately the age of 4 years. All the subjects were Italian and had a positive anti-VZV ELISA test result (Table 2).

According to the questionnaire given at the time of enrolment, 42/165 recalled having had varicella, which was confirmed by their positivity to the serological test. However, the varicella notification had been recorded for only six of them. 

### 3.3. Vaccination Status against Varicella and Seroprevalence Assessment

According to the regional vaccination registry, 61.2% of subjects were vaccinated (101/165) against varicella, and the vaccine administered was tetravalent MMRV or the monovalent (V). Most of the vaccinated subjects were found to have a positive response to the anti-VZV (73.3%), while 10.9% of subjects were negative and the remaining 15.8% had an equivocal result. In the vaccinated population, all the subjects belonging to the 15–18 years age group tested positive in response to anti-VZV; in the other three age groups, the percentages of seropositivity tended to gradually decrease from the younger subjects (74.3% in subjects aged 1–4 years) to the older ones (65% in subjects aged 10–14 years). The percentage of negative subjects had an opposite trend in these same age groups, and the equivocal sera ranged between 15.4 and 20% (Table 3). The unvaccinated group contained 64 subjects. Most of them (79.7% 51/64) tested positive in response to the anti-VZV test; 17.2% of sera samples (11/64) were negative and 3.1% were equivocal (2/64). All the subjects aged 1–4 years tested negative. The seropositivity percentage, instead, tended to increase among the older subjects, from 55.6% in the 5–9 years age group up to 95% in the 15–18-year-old subjects. Susceptible subjects (unvaccinated and negative) gradually decreased in the older subjects, from 22.2% in the 5–9 years group to 5% in the 15–18 years group. Equivocal sera were found only in the 5–9-year-old subjects (22.2%) (Table 3).

### 3.4. Seroprevalence Distribution Related to the Number of Vaccine Doses

We also retrieved the number of vaccine doses received by each subject (Figure 2). Among the vaccinated population, 57.4% (58/101) of subjects received one dose and 42.6% (43/101) received two doses. Higher seroprevalence was found in subjects who had received two doses (81.4%) than those who had received just one dose (67.2%). The percentages of negative sera and equivocal sera were lower in subjects vaccinated with two doses (9.3% for both negative and equivocal sera) when compared with those who had received one dose of vaccine (12.1% and 20.7%, respectively). Nevertheless, no significant differences in seropositivity were found among the vaccinated subjects when comparing those who received one dose with those who received two doses (*p* > 0.05).

The overall seroprevalence distribution related to the time elapsed since the last dose of vaccine received highlighted the fact that the seropositivity lasted over time, up to 12 years; no significant trend was indeed found assessing a possible antibody decline in those who were vaccinated years before their enrolment in this study (*p* > 0.05). Vaccinated negative subjects were distributed over time, and the percentage was lower than the positive ones. The distribution of equivocal sera samples tended to be more consistent after 1–3 years since the vaccination. For one subject, we were unable to retrieve the year of the last dose received, but the serum sample was found positive (Figure 3).

## 4. Discussion

The aim of this study was to obtain a picture of the level of varicella immunity and susceptibility in a pediatric and adolescent population of the Province of Florence. Furthermore, we assessed whether the anti-varicella immunization was due to vaccination or to natural infection. Our study population well represented the general Italian scenario: the non-Italian citizen percentage in our sample was 17%, and about 20% of children in Italy were born from at least one foreign parent as of 2018 [23]. Our data showed an overall varicella seroprevalence of 75.8% (24.2% among negative and equivocal subjects). Interestingly, the seropositivity trend significantly increases with age: 65% in 1–4-year-olds, 68.7% in 6–9-year-olds, 80% in 10–14-year-olds and 96.3% in 15–18-year-olds. Therefore, the number of susceptible people (negative and equivocal) shows an inverse trend with age. Thanks to the UVV program (set up in 2008 in Tuscany), the varicella cases successfully dropped from 350 notified cases/100,000 in 1992 to 20 notified cases/100,000 in 2018 [17]. Unlike the other seroprevalence studies carried out in the same sample of measles [19], rubella [20] and hepatitis B [21], in which no cases were reported to the surveillance system of infectious diseases, seven notifications were observed for varicella. Since varicella infections are rarely asymptomatic, the clinical diagnosis is generally easy [24]. Indeed, according to our results, we can estimate a certain degree of underreporting (1:6) for varicella notifications, as 42 subjects in our study population reported a natural history of varicella disease (by parent/patient recall), and a positive history of varicella is a reliable marker of disease [25]. The average age of the notified cases in our sample is 4 years, which is a lower value than the median age in Tuscany (9 years in 2018) [17]. Besides the established two-dose vaccination schedule, the Tuscany region offers the anti-VZV vaccination to adolescents aged 11–14 years who have not contracted varicella previously, recommending the administration of the monovalent vaccine with a two-dose schedule [15,26]. We assessed the vaccination registry database and found 14 subjects recorded as “protected”: 11 of them were aged ≥10 years old, while one subject was 7 years old and the last two aged 9 years old. We could hypothesize that they were tested for anti-VZV antibodies and, resulting to be positive, they have not received the vaccination. Our serological results confirmed positive results in response to anti-VZV antibodies for 13 of these subjects, and just 1 of them appeared to be weakly immunized, as their serum had an equivocal outcome. No “protected” subject was found in the “unvaccinated–protected” 1–4 years age group. Considering the positive serological result, the “unvaccinated–protected” subjects were 22.2% in the 5–9 years age group, 16.7% among the 10–14 years age group and 30% among the oldest age group (15–18 years). However, the highest seropositivity percentages were measured among the “unvaccinated–unprotected” subjects in all the different age groups. Notably, all the unvaccinated and unprotected subjects aged 1–4 years were negative. Although subjects who received more than one dose showed higher seropositivity than those who received only one (81.4% vs. 67.2%), this was not statistically significant. This may be due to the small size of our vaccinated population: 58/101 received one dose and 43/101 received two doses. Despite the high efficacy of the vaccine, serological studies suggest a significant primary vaccine failure after one dose of varicella vaccine in the United States [27,28,29,30]. Because of school outbreaks and continued transmission of wild-type VZV from vaccinated subjects to others, a second dose of varicella vaccine for all children was recommended by the Centers for Disease Control and Prevention (CDC) in June 2006 [31], boosting the varicella seropositivity after the second dose. Moreover, in our study, anti-varicella immunity protection was confirmed to be long-lasting, persisting up to 12 years since the last vaccine administration [32,33]. 

## 5. Conclusions

Our seroprevalence study gives an accurate picture of the level of anti-varicella immunization in 1–18-year-old subjects residing in the province of Florence (Italy). Although varicella seroprevalence is below 95% in almost all the age groups of this study population (except for the 15–18-year-old subjects), our data are encouraging and reflect the success of the introduction of the UVV program and the vaccination campaigns promoted in the region. Varicella is still a highly notifiable disease, especially in young children. Therefore, it is crucial not only to maintain the disease notification system and surveillance at a good level, but also to monitor the vaccination administration constantly and efficiently to improve the VC and herd immunity. 

## Figures and Tables

**Figure 1 vaccines-09-00152-f001:**
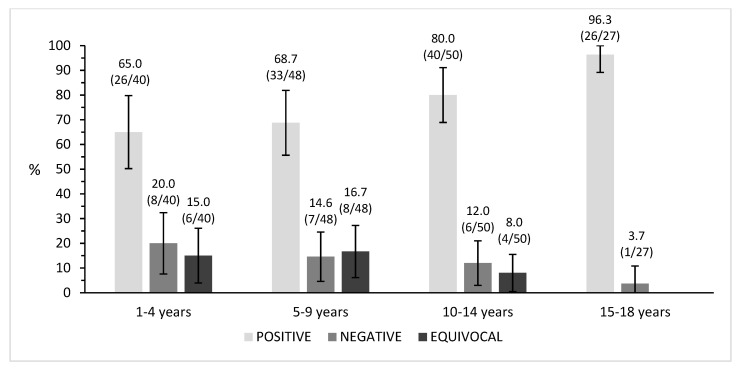
Percentage of immunity distribution in the age groups. Percentages and ratios (*n*/*N*) are shown above each column.

**Figure 2 vaccines-09-00152-f002:**
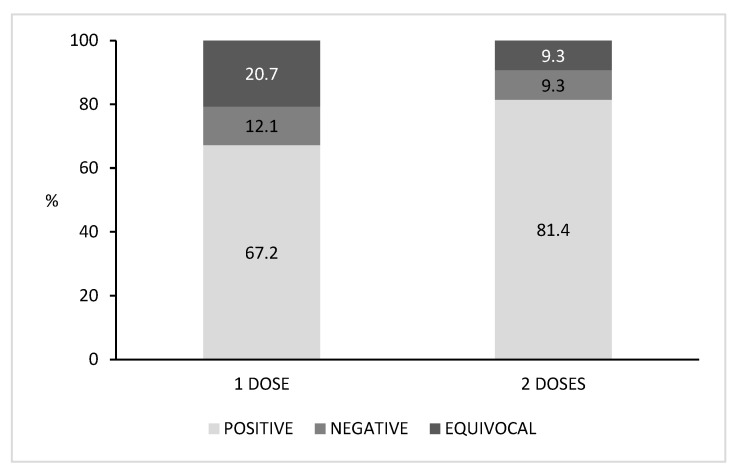
Percentage distribution of positive, negative and equivocal subjects according to the number of vaccine doses received.

**Figure 3 vaccines-09-00152-f003:**
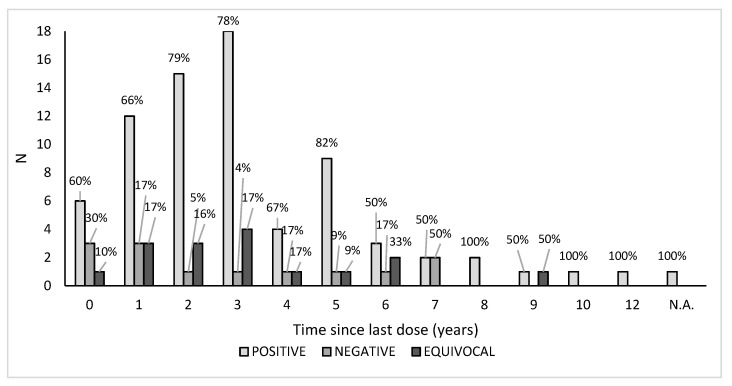
Seroprevalence related to the time (years) elapsed since the last dose of vaccine received. N.A.: not available.

**Table 1 vaccines-09-00152-t001:** Anti-varicella seroprevalence (*n*/*N*: number of subjects/total).

Anti-VZV Seroprevalence			
Group	Positive % (*n*/*N*)	Negative % (*n*/*N*)	Equivocal % (*n*/*N*)
Male	75.9 (66/87)	14.9 (13/87)	9.2 (8/87)
Female	75.6 (59/78)	11.5 (9/78)	12.8 (10/78)
Italian	75.8 (113/149)	12.8 (19/149)	11.4 (17/149)
Non-Italian	75.0 (12/16)	18.8 (3/16)	6.3 (1/16)
Overall	75.8 (125/165)	13.3 (22/165)	10.9 (18/165)

**Table 2 vaccines-09-00152-t002:** Notification of varicella disease according to The National Registry of Notifications for Infectious Diseases (SIMI).

Year of Birth	Age at the Time of Enrolment (Years)	Sex	Notification (Year)	Age at the Time of Disease Notification (Years)
2000	17	F	2006	6
2002	16	M	2006	4
2002	15	M	2006	4
2004	14	F	2008	4
2006	11	M	2010	4
2007	10	M	2011	4
2012	5	F	2015	3

**Table 3 vaccines-09-00152-t003:** Vaccination status of the enrolled subjects related to the serological result (n/N: number of subjects/total).

Vaccination Status	Age Group (years)	Positive % (*n*/*N*)	Negative % (*n*/*N*)	Equivocal % (*n*/*N*)	Total % (*n*/*N*)
Vaccinated		73.3 (74/101)	10.9 (11/101)	15.8 (16/101)	61.2 (101/165)
	1–4	74.3 (26/35)	8.6 (3/35)	17.1 (6/35)	34.7 (35/101)
	5–9	71.8 (28/39)	12.8 (5/39)	15.4 (6/39)	38.6 (39/101)
	10–14	65.0 (13/20)	15.0 (3/20)	20.0 (4/20)	19.8 (20/101)
	15–18	100.0 (7/7)	0.0 (0/7)	0.0 (0/7)	6.9 (7/101)
Unvaccinated		79.7 (51/64)	17.2 (11/64)	3.1 (2/64)	38.8 (64/165)
	1–4	0.0 (0/5)	100.0 (5/5)	0.0 (0/5)	7.8 (5/64)
	5–9	55.6 (5/9)	22.2 (2/9)	22.2 (2/9)	14.1 (9/64)
	10–14	90.0 (27/30)	10.0 (3/10)	0.0 (0/29)	46.9 (30/64)
	15–18	95.0 (19/20)	5.0 (1/20)	0.0 (0/20)	31.2 (20/64)

## Data Availability

Data sharing not applicable. Data were collected and managed in aggregated form according to the European Union Regulation 2016/679 of the European Parliament and the Italian Legislative Decree 2018/101.

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
