# Peer review of "A Study of Varicella Seroprevalence in a Pediatric and Adolescent Population in Florence (Italy). Natural Infection and Vaccination-Acquired Immunization"

_vaccines, 2021, doi:10.3390/vaccines9020152_

Round 1

Reviewer 1 Report

Line 23: You mention in your abstract: "Varicella is a well-known infectious disease, which may have severe complications, especially in young children." Please correct that severe complications do not "especially" occur in young children but rather "also in young children" - other risks groups are adults, immunocompromised and newborns of women having a rash between 5 days 52 before and 2 days after delivery, as you mention in your introduction. 

Line 60: "deceased" sounds a bit strange in this context. I would recommend using "death" or "fatal cases"

Line 63: If I am not mistaken, one-dose vaccination was introduced in the United States in 1995, not 1996? Please add reference, for instance: https://www.cdc.gov/mmrw/pdf/rr/rr5604.pdf). (Note that link in reference 3 - lines 315/316 - does not seem to lead anywhere!)

Lines 64-65: Please correct the definition of "breakthrough vaccination" as a case of infection with wild-type VZV occurring >42 days after vaccination

Lines 72-74: Please rephrase and correct the English: "A two-dose schedule, with an interval between 13th-15th month and 5-6 years old, is used to administered Measles-Mumps-Rubella-Varicella (MMRV) vaccines in most countries in Europe" - I suggest: "A two-dose Measles-Mumps-Rubella-Varicella (MMRV) vaccination schedule is generally administered around 13-15 months and 5-6 years old in most countries in Europe."

Line 77: I suggest to reword as: "This program has since been expanded to the whole country with the “National Immunization Plan (NIP) 2017-2019”

Lines 83-86: For a better understanding, I suggest rewording and correcting the English as such: "Even though the number of varicella notifications in Tuscany have dramatically fallen in the last decades, the notification rate remains high (from ~350 cases/100.000 in 1994 to 20 cases/100.000 in 2018 - i.e > 700 notifications)"

Lines 86-87: I suggest to correct as such: "...the age at which the disease is contracted is..."

Lines 90-92: Please clarify as of when this surveillance system has been set up?

Line 163: Please clarify or correct: "...and 27 in the age group 1-18 years old": do you rather mean in the age group 15-18?

Line 194-195: I suggest to rephrase as such: "However, the disease was only notified for six of them."

Lines 278-280: please elaborate and reconcile your Discussion comment "Looking at the number of vaccine doses, subjects who received more than one show a higher sero-positivity (81.4% with two doses vs 67.2% with one dose)", with your Data Analysis observation described in lines 223-225 ("Nevertheless, no significant differences in seropositivity were found among the vaccinated subjects, when comparing those who received 1 dose with those who received 2 doses (p>0.05)."): is this due to a sample size that is too limited?

Author Response

Comment n. 1: Line 23: You mention in your abstract: "Varicella is a well-known infectious disease, which may have severe complications, especially in young children." Please correct that severe complications do not "especially" occur in young children but rather "also in young children" - other risks groups are adults, immunocompromised and newborns of women having a rash between 5 days 52 before and 2 days after delivery, as you mention in your introduction.
Response n.2: Thank you for your comment. We corrected as you suggested.
Comment n. 2: Line 60: "deceased" sounds a bit strange in this context. I would recommend using "death" or "fatal cases"
Response n.2: We corrected “deceased” with “deaths”.
Comment n. 3: Line 63: If I am not mistaken, one-dose vaccination was introduced in the United States in 1995, not 1996? Please add reference, for instance: https://www.cdc.gov/mmrw/pdf/rr/rr5604.pdf). (Note that link in reference 3 - lines 315/316 - does not seem to lead anywhere!)
Response n. 3: Thanks for your suggestion. We confirm that one-dose vaccination was introduced in the United States in 1995. We changed the years in the manuscript and the reference n. 3 with the one which the Reviewer has suggested.
Comment n. 4: Lines 64-65: Please correct the definition of "breakthrough vaccination" as a case of infection with wild-type VZV occurring >42 days after vaccination
Response n.4: Thank you for your comment. We changed the definition as suggested.
Comment n. 5: Lines 72-74: Please rephrase and correct the English: "A two-dose schedule, with an interval between 13th-15th month and 5-6 years old, is used to administered Measles-Mumps-Rubella-Varicella (MMRV) vaccines in most countries in Europe" - I suggest: "A two-dose Measles-Mumps-Rubella-
Varicella (MMRV) vaccination schedule is generally administered around 13-15 months and 5-6 years old in most countries in Europe."
Response n.5: We corrected as you suggested.
Comment n. 6: Line 77: I suggest to reword as: "This program has since been expanded to the whole country with the “National Immunization Plan (NIP) 2017-2019”
Response n.6: Thank you. We modified the sentence as suggested.
Comment n. 7: Lines 83-86: For a better understanding, I suggest rewording and correcting the English as such: "Even though the number of varicella notifications in Tuscany have dramatically fallen in the last decades, the notification rate remains high (from ~350 cases/100.000 in 1994 to 20 cases/100.000 in 2018 - i.e > 700 notifications)"
Response n.7: Sentence was modified as suggested.
Comment n. 8: Lines 86-87: I suggest to correct as such: "...the age at which the disease is contracted is..."
Response n.8: Sentence was modified as suggested.
Comment n. 9: Lines 90-92: Please clarify as of when this surveillance system has been set up?
Response n.9: Modified as: “...has been established since January 2000”.
Comment n. 10: Line 163: Please clarify or correct: "...and 27 in the age group 1-18 years old": do you rather mean in the age group 15-18?
Response n.10: We made a typos mistake: we meant to type 15-18 years old. We changed it in the manuscript.
Comment n. 11: Line 194-195: I suggest to rephrase as such: "However, the disease was only notified for six of them."
Response n.11: Thank you. We modified the sentence as suggested.
Comment n. 12: Lines 278-280: please elaborate and reconcile your Discussion comment "Looking at the number of vaccine doses, subjects who received more than one show a higher sero-positivity (81.4% with two doses vs 67.2% with one dose)", with your Data Analysis observation described in lines 223-225 ("Nevertheless, no significant differences in seropositivity were found among the vaccinated subjects, when comparing those who received 1 dose with those who received 2 doses (p>0.05)."): is this due to a sample size that is too limited?
Response n.12: This may be due to the sample size, which is small: among 101 vaccinated subjects, 58 of them received one dose whereas 43 received 2 doses. In order to clarify this point, we added these numbers in the text (lines: 221-222) and we rephrased the sentence at lines 281-285 as follows: “Although subjects who received more than one dose show a higher seropositivity than those who received only one (81.4% vs 67.2%), this was not statistically significant. This may be due to the small size of our vaccinated population: 58/101 received one dose and 43/101 received two doses”.

Reviewer 2 Report

Is is a retrospective observational study of regional varicella seroprevalence. 

---- General

It is a part of a larger vaccination surveillance project and has been pre-registered = definite strengths of the work.

The study very nicely shows the successful execution of a regional vaccination program and produces encouraging results that prove the effectiveness of vaccines. 

The conclusions are reflecting the findings described in the paper and are not overstated. 

---- Major issues
No major comments

---- Minor issues
line 60 substitute deceases with deaths
line 89 (and many more) change 'notification disease' to 'notifiable disease'
The manuscript requires a thorough language revision. 

Author Response

Comments and Suggestions for Authors
Is is a retrospective observational study of regional varicella seroprevalence.
---- General
It is a part of a larger vaccination surveillance project and has been pre-registered = definite strengths of the work.
The study very nicely shows the successful execution of a regional vaccination program and produces encouraging results that prove the effectiveness of vaccines.
The conclusions are reflecting the findings described in the paper and are not overstated.
Comment n. 1: line 60 substitute deceases with deaths
Response n.1: Thank you for your suggestion. We changed “deceases” with “deaths”.
Comment n. 2: line 89 (and many more) change 'notification disease' to 'notifiable disease'
Response n.2: We changed that according to the Reviewer’s suggestion. We changed 'notification disease' to 'notifiable disease' in line 89. However, we kept the other "notifications" when appropriated because they were referred to the system (notification system) or (number of) notifications.